# The Impact of Social Pension Schemes on the Mental Health of the Chinese Elderly: A Mediating Effect Perspective of Two-Way Intergenerational Support

**DOI:** 10.3390/ijerph19148721

**Published:** 2022-07-18

**Authors:** Dongling Zhang, Yanyan Wang, Yuxin Jiao

**Affiliations:** School of Economics, Qingdao University, Qingdao 266061, China; zhangdongling99@126.com (D.Z.); 2020025057@qdu.edu.cn (Y.J.)

**Keywords:** pension schemes, mental health of elderly, two-way intergenerational support, reverse intergenerational transfer, quantile regression

## Abstract

With the rapid decline in China’s fertility rate, the acceleration of aging, the continuous miniaturization and nucleation of China’s family structure, and the deterioration of the elderly’s living conditions and mental states, the elderly’s mental health has emerged as a major public health issue and a major social problem. Pensions are the elderly’s primary source of income, and they can help them meet their basic needs while also promoting family harmony and improving their mental health. Based on the data of the 2012, 2016, and 2018 Chinese Family Panel Studies (CFPS), we empirically examined the impact of pensions on the elderly’s mental health by using the fixed effects model, mediating effect model, and so on. The results show that receiving pension benefits can significantly reduce the level of depression and improve the mental health of the elderly. Receiving pension benefits causes reverse intergenerational economic transfer, which reduces the effect of pensions on the elderly’s mental health to some extent, but the life care and mental comfort provided by children increases when the elderly share pensions with their children. Overall, receiving pension benefits improves the mental health of the elderly. In addition, the effect of pensions on the elderly with different characteristics is heterogeneous. The older elderly, women, the elderly living with their children, and the elderly in rural areas all benefit more from receiving pension benefits. Moreover, the worse the mental health of the elderly is, the stronger the influence of pensions on their mental health is. Our discussion has important policy implications.

## 1. Introduction

As China enters a new phase of development, its population structure has taken on new characteristics, and the aging trend cannot be overlooked. According to the data of the Seventh Census in China conducted in 2020, the population aged 65 and up in China has reached 190.64 million, occupying 13.5 percent of the total population. In comparison to the data from the Sixth Census in China conducted in 2010, China’s total population aged 65 and up increased by 71.7 million, and the proportion of the elderly population increased by 4.63 percentage points [1]. According to the Population Division of the UN’s Department of Economic and Social Affairs’ 2019 World Population Prospects Report, China’s population aging rate is anticipated to be 20.7 percent in 2035 and 26.1 percent in 2050, placing it 33rd in the world [2]. The elderly’s organs steadily age, numerous functions such as immunological function deteriorate, social resources are depleted, and social relationships are impaired as they grow older. In addition, as the family structure becomes more centralized, the traditional family model of aging also weakens, causing loneliness, depression, and other negative emotions in the elderly, all of which have an impact on their mental health. According to the National Health Commission of the People’s Republic of China, just 30.3 percent of the urban elderly and 26.8% of the rural elderly in China are in good mental health [3]. It is clear that China’s elderly population’s mental health is not in good shape. In the future, improving the mental health of the elderly will be the key to actively responding to aging and achieving the strategic goal of healthy China.

China has joined the ranks of upper-middle-income nations, but its economic and social growth are still problematic, and its public service system is not yet sound. However, China has reached the low fertility and low mortality levels seen in high-income countries. People’s desire to have children is waning, and population growth is slowing considerably, resulting in a sharp drop in the number of children and an acceleration of the aging trend [4]. With fewer children and more older members in the family, the family’s old-age support ratio falls, and the family’s intergenerational support load rises [5]. Furthermore, China’s long-term absence of public service policies for the elderly, such as life care and spiritual comfort, has resulted in the deterioration of the elderly’s living conditions and mental state. Many elderly people suffer from serious psychological issues, and mental illness is a major risk factor for suicide [6]. The Chinese government hopes to provide basic living security for the elderly by optimizing the income distribution pattern and providing income maintenance programs to protect vulnerable groups of the elderly, promote family harmony, and achieve the goal of “mental health with old age, peace, and happiness in old age”.

In September 2009, the China State Council promulgated the “Guidance Opinions on Piloting the New Rural Pension Scheme”, and the New Rural Social Pension Scheme (NRSP) assumed responsibility for the pilot program’s implementation in 320 counties across the country. The NRSP was quickly implemented across the country. By the end of 2012, all 2853 counties (cities and districts) in China had embraced the NPSP, and 460 million people were covered [7]. It is stipulated in the policy that rural residents who have reached the age of 16 but have not joined the Urban Employees’ Basic Endowment Insurance System (students in school excluded) are eligible to enroll in the NRSP in their hometown. After the implementation of the NRSP, individuals who had reached the age of 60 and had not received the basic pension benefits of urban workers were eligible for a free monthly basic pension, while those who had not reached the age of 60 were obliged to pay the premium annually. After the implementation of the NRSP pilot program, the China State Council launched a national pilot program of the Urban Residents’ Pension Scheme (URSP) in July 2011 to expedite the development of a social security system covering urban and rural residents. In terms of the system model, pension computation and payment mechanism, financial subsidy form, and personal account and fund administration, URSP and NRSP are similar. However, in contrast to the NRSP, the URSP’s participants are 16-year-old urban non-working residents (students in school excluded) who do not fit the criteria for worker pension systems. Early in 2014, the China State Council integrated the NRSP and the URSP, establishing a unified social pension scheme for urban and rural residents, thus eliminating the divide between urban and rural pension schemes. The policy framework for the integrated pension scheme is still the same, except that the base pension amount and payment level have been changed.

As the primary source of income for the elderly, social pension benefits are a critical step in mitigating and preventing dangers to the elderly, which can have a substantial impact on their mental health. On the one hand, pension benefits can increase the economic income of the elderly, help them to live a dignified life, and diminish their psychological sense of deprivation [8]. On the other hand, pension benefits can also influence the intergenerational economic support that the elderly receive. Scholars have discovered that pension benefits have a “crowding out” effect on intergenerational economic support, which will replace the transfer payments from children, with children being the largest beneficiaries [9]. The “crowding out” effect may diminish the effectiveness of the pension scheme. However, according to some research, pension-eligible seniors can utilize a portion of their pensions to compensate their children in order to entice them to offer the elderly additional care and spiritual comfort [10]. This helps to reduce loneliness among the elderly. Can receiving pension benefits truly alleviate depression and improve the elderly’s mental health? What role does intergenerational support play in mediating the relationship between pension schemes and the elderly’s mental health? The purpose of our paper is to conduct an in-depth empirical analysis of the impact of pensions on the elderly’s mental health and the mediating effect of intergenerational support on the relationship between pension benefits and the elderly’s mental health, based on data from the Chinese Family Panel Studies (CFPS), in the macro background of weakened intergenerational family support and low pension benefit level. Additionally, we examine whether the influence of pension benefits on the elderly’s mental health varies depending on their characteristics. Analysis of the aforementioned issues can not only serve as a theoretical and logical complement to existing academic research on the impact of pensions on the elderly’s mental health, but also as a scientific foundation for decision making to optimize the allocation of family and social pension resources and improve the elderly’s mental health.

## 2. Literature Review and Research Hypothesis

### 2.1. Literature Review

Currently, research on the elements that influence older mental health is mostly focused on three aspects. First, demographic factors such as the elderly’s age and gender, as well as their physical ailments, have an impact on their mental health. For example, Li et al. [11] discovered that women had a greater overall prevalence of depressive symptoms than males and that depression in the elderly with living spouses was much lower than depression in the elderly who were single. Furthermore, as education levels improved, the prevalence of elderly depression symptoms decreased. Guerra et al. [12] examined data from nine low- and middle-income nations, including China, India, and Cuba, and discovered that as people become older, their depression symptoms worsen, and women’s depression symptoms are higher than men’s. Qin et al. [13] discovered that mental health has a considerable socio-economic gradient in addition to age, gender, and regional features. Higher levels of education and money were linked to a lower risk of depression. Freemen et al. [14] discovered that those with a low socioeconomic status had higher rates of depression and poor mental health. Susanty et al. [15] discovered that the elderly’s physical health has an impact on their loneliness. The elderly’s loneliness would be exacerbated by poor health, chronic sickness, and limited cognitive ability.

Second, social activity has an impact on the elderly’s mental health. Bruce and Hoff [16] discovered that social isolation and inactivity enhance the likelihood of experiencing serious depression for the first time using data from the New Haven Epidemiologic Catchment Area. Lower levels of outdoor and recreational exercise were linked to an increased risk of depression four years later, according to Morgan [17]. Chi et al. [18] found that social participation benefits the elderly’s psychological health in general, and that continuing or launching social activities, in particular, reduces depressive symptoms that are likely to occur later in life. After controlling for age, sex, time, education, marital status, health, functional status, and fitness activities, Glass et al. [19] found that social interaction is associated with lower CES-D scores. According to Wang et al. [20], different social activities had diverse effects on depression in the elderly. Friendship, exercise, and recreational activities were found to be useful in reducing the incidence of depression in the elderly when sample selection bias was taken into account. In the elderly, sports and recreational participation were linked to a lower risk of depression than other social activities.

Finally, intergenerational family support has an impact on the mental health of the elderly. Most experts have argued that receiving intergenerational support from their offspring lowers the likelihood of depression in older people. Lee and Xiao [21] discovered that children’s daily care for the elderly can improve their feelings of authority and reciprocity, both of which are beneficial to their mental health. Zhang and Li [22] discovered that emotional communication in intergenerational support has a significant impact on the elderly’s mental health. Intergenerational support from sons is very crucial in preserving and strengthening the mental health of the elderly. Intergenerational economic support, housework support, daily living assistance, and intergenerational emotional support, according to Wang and Li [23], were all beneficial to the elderly’s mental health development. Nie [24] found that the number of children and the contentment of the elderly in rural areas had no significant link, and the primary elements influencing the subjective happiness of the elderly in rural areas were financial support and life care for their children. However, some researchers believed that young people assisting the elderly increases the elderly’s risk of depression. Dean et al. [25] discovered that receiving their children’s help made the elderly feel old and increased their reliance on others. The elderly’s depression level was positively connected with the frequency and intensity of receiving their children’s assistance. Krause et al. [26] discovered that the elderly who are unable to care for themselves require 24 h care. Children who provide long-term care to the elderly experience a high degree of unpleasant emotions, which can quickly lead to intergenerational disputes and an increase in the elderly’s despair. Furthermore, government subsidies and environmental factors can have an impact on the elderly’s mental health [27,28,29,30].

There are now two opposing opinions on the impact of pensions on the elderly’s mental health. The first is that pensions can significantly improve the elderly’s mental health. Inadequate pensions, according to Adler et al. [8], would affect the elderly’s mental health, as a lack of living resources would reduce the elderly’s ability to handle stressful life events, resulting in depression, hostility, and psychological tension, as well as psychological and emotional deprivation, but these problems would not appear when pensions are adequate. Sagner et al. and Case et al. [31,32] investigated pension schemes in Africa and discovered that by sharing their pensions with their families, the elderly in Africa acquired pride and respect from their family members. Although pensions would crowd out intergenerational economic support for the elderly, Kohil et al. [10] observed that when pension increases limit cash support for children, spiritual comfort and emotional support supplied by offspring may increase, boosting the total welfare of the elderly. Case and Zhang [28,33] believed that having pensions could significantly reduce the anxiety and loneliness of the elderly and improve their life satisfaction. Life satisfaction can reflect the general feeling of life [34]. Although there are some differences between life satisfaction and negative emotions such as depression, the two are strongly linked [35]. From the perspective of multidimensional poverty among the elderly, Liu [36] discovered that taking out social endowment insurance can significantly reduce the likelihood of the elderly falling into mental poverty. Receiving the New Rural Social Pension significantly improved the mental health of the elderly in rural areas, according to Zhou et al. [37]. The marginal effect of the pensions was stronger for rural women and people with lower economic status. According to Pan et al. [38], pension schemes not only reduced depressive symptoms in rural residents but also kept the prevalence rate of depression low. When it comes to depression relief, pensioners benefited more than contributors from joining the pension schemes.

The second point of view is that the impact of pensions on elderly mental health is unclear. According to Lv [39], the elderly’s mental health index was greater under the family support mode than under the social support mode. Xie [40] believed that China’s New Rural Social Pension had a low benefit level and that it struggled to affect the welfare of the elderly in the short term, and so it had no significant impact on the elderly’s depression and mental health. The social endowment insurance pension had no significant impact on the mental health of the elderly and could not reduce the likelihood of the elderly falling into mental poverty, according to Zhu [41]. Guo [42] found that the influence of social security on the health of the elderly in rural areas is weak, and the impact of the New Rural Social Pension on the elderly’s subjective mental health is unclear. Based on a multidimensional view of poverty, Liu et al. [43] discovered that the New Rural Social Pension can help the elderly improve their income and reduce their income poverty while having a minor impact on health and psychological poverty. It is clear that no consensus exists on the influence of pensions on the elderly’s mental health.

According to a study of available studies, the views of academics are divided on how pensions impact the mental health of the elderly, with the majority of scholars affirming that pensions have a favorable effect on the elderly’s mental health. However, although some scholars, based on the perspective of multidimensional poverty, have explored the impact of China’s pension plan on the psychological or spiritual poverty of the elderly, their psychological status is only measured by the loneliness of a single dimension of the elderly. Furthermore, most Chinese researchers have looked into the impact of intergenerational support on the mental health of the elderly, but they have not considered whether pensions affect the direction of intergenerational support and then affect the mental health of the elderly. However, as China’s social security systems improve, they have become an increasingly vital force in supporting the aged. As a result, it is vital to investigate the impact of pension schemes on the elderly’s mental health. Based on this, we examine the influence of China’s pension schemes on the elderly’s mental health as well as whether the pensions will affect the elderly’s mental health through the impact of intergenerational family support, using data from the Chinese Family Panel Studies.

### 2.2. Research Hypothesis

Pensions, as a “safety net” for the elderly, provide them with steady and consistent economic resources and ensure that they can obtain their basic necessities for living, thus having a significant impact on their mental health. On the one hand, pensions can improve the relative economic status of the elderly inside and outside the family, reduce the economic dependence of the elderly on their children, reduce the psychological and emotional sense of deprivation when the elderly cannot cope with emergencies due to economic constraints, and help the elderly to gain self-esteem and social respect [44]. On the other hand, some scholars found that pensions not only have a “crowding out” effect on intergenerational economic support, but in some parts of China, the elderly pay for their children’s pension premiums to receive the basic pension. The pension is indirectly transferred from the old to adults, resulting in a “reverse income redistribution” phenomenon [45]. This may lessen the impact of the pension schemes on the elderly’s mental health. The health economics hypothesis holds that the mental health of parents is primarily determined by the factor input of productive mental health [46]. Intergenerational support, particularly children’s spiritual comfort, is a significant and irreplaceable input factor for parents’ mental health in East Asia, which is heavily impacted by Confucian culture [47,48]. Pensions can be used as a legacy or reward for the elderly to “purchase” services from their offspring that are not available on the market [32,49]. The pensions increase the total amount of care provided by the family to the elderly, reducing loneliness and anxiety and improving the elderly’s mental health. In other words, when older people share their pensions with their offspring, the amount of intergenerational life care and spiritual comfort they receive rises. Overall, receiving pension benefits improves elderly people’s mental health. As a result, pensions can have an impact on the mental health of the elderly, not just directly but also indirectly through intergenerational support. Therefore, we propose the following hypotheses:

**Hypothesis** **1** **(H1).**
*Pensions can improve the mental health level of the elderly.*


**Hypothesis** **2** **(H2).**
*Pensions can affect mental health in old age by influencing intergenerational family support.*


## 3. Materials and Methods

### 3.1. Data Source and Sample Selection

The data in this article came from Peking University’s Social Science Survey Center’s 2012–2018 Chinese Family Panel Studies (CFPS). These surveys covered all family members in 16,000 households throughout 25 provinces, autonomous areas, and municipalities under the control of the Chinese government. These surveys focused on economic activities, educational success, family relationships, migration, and family health problems at the individual, household, and community levels. We processed the data as follows, based on the research needs: to begin with, the adult and family questionnaires were matched and joined. Because the survey data from 2014 contained some missing CES-D scale data, the data from that year were omitted, leaving only the survey data from 2012, 2016, and 2018. Second, considering that the beginning age of eligibility for a pension is 60, only samples aged 60 and older were included in this study, while samples temporarily below the beginning age of eligibility for a pension were eliminated. Additionally, we also put the data together into balanced panel data, which gave us a total of 9672 sample observations.

### 3.2. Variables Selection

#### 3.2.1. Explained Variable

The level of depression measured by the CES-D scale is used to measure mental health in studies across multiple disciplines [50,51]. Radloff came up with this scale in 1977 [52]. The full scale has 20 items, 16 of which are negative emotions and four are positive emotions. The frequency of events in the previous week is used to rate each item. Differing survey years have different CES-D scale measurements in the Chinese Family Panel Studies (CFPS) database. In 2018 and 2012, the CES-D20 scale was primarily used. In the 2016 survey, 1/5 of the questioned population was randomly chosen to use the full version of the CES-D20 scale, while the remaining 4/5 samples used the CES-D8 with an 8-item brief scale. In the 2014 survey, the psychological health status of respondents was mainly investigated through the monthly frequency of six items.

Referring to the study of Xi [53], we used the CES-D short form with 8 items to measure the health status of the elderly. The scale’s reliability and validity were confirmed by Turvey et al. [54]. According to the frequency of occurrence per week, each item was given a score of 1–4. Finally, the above questions were added together, and the positive emotion questions were transposed. The higher the score, the more depression there is, and the worse the elderly’s mental health is.

#### 3.2.2. Explanatory Variable

Whether the elderly receive the social pension scheme was the main explanatory variable in this study. This variable was set as 1 if the elderly person received pension benefits, and 0 otherwise. The social endowment insurance pensions included both NRSP and URSP, and the samples that receive pensions other than NRSP and URSP were deleted to avoid interference with the accuracy of the estimation results.

#### 3.2.3. Mediating Variables

The mediating variables in this study were reverse intergenerational economic support, intergenerational life care, and intergenerational spiritual comfort. These three factors are well-representative and can accurately assess the situation of intergenerational support. The reverse intergenerational economic support was calculated using the difference between the amount of economic assistance provided by the elderly to their children and the amount of intergenerational economic support received by the elderly from their children, i.e., the reverse intergenerational transfer amount provided by the elderly to their children, and further logarithmic processing was performed. Intergenerational life care is measured by the frequency of children helping parents with household chores and caring for the elderly. According to the frequency of occurrence per month, it is given a score of 1–6. The greater the value is, the more intergenerational care the children provide for their parents. Intergenerational spiritual comfort is measured by the degree of intimacy between old people and their children. According to the degree of intimacy between old people and their children, it was given a score of 1–5. The greater the value, the tighter the bond between the elderly and their children, the greater the depth of feeling, and the greater the spiritual comfort the elderly receive.

#### 3.2.4. Control Variables

Based on the existing literature, we selected two variables of the elderly’s characteristics and family characteristics as control variables, which include age, gender, marital status, education level, and physical health status. The variables of family characteristics include per capita net income, family size, and household registration.

### 3.3. Research Methods

#### 3.3.1. Basic Model

There is a significant discrepancy in the samples in terms of educational background and family economic status, as well as the macroeconomic environment and policies of the samples during each survey period. To assure the study’s rigor, the bidirectional fixed-effect model was used to control the individual fixed effect and time-point fixed effect of samples to examine the effects of receiving pension benefits on the mental health of the elderly. The basic model was built as follows:(1)$$Y_{it}=\alpha +\beta X_{it}+\delta Z+\mu_{i}+\eta_{t}+\varepsilon_{it}$$
where $Y_{it}\;$ represents the CES-D score of the elderly, $X_{it}^{\;}\;$ represents whether the elderly get pensions, *Z* represents the control variable, $\mu_{i}$ represents the individual fixed effect,$\;\eta_{t}$ represents the year fixed effect, and $\;\varepsilon_{it}$ represents the random disturbance term.

#### 3.3.2. Mediating Effect Model 

The stepwise regression method was used to test whether pensions can affect the mental health of the elderly by affecting intergenerational support. In accordance with the research of relevant scholars [55,56,57], relevant models were set as follows:(2)$$Y=\beta_{0}+\beta_{1}X+\beta_{2}Z+\varepsilon_{1}$$
(3)$$M=\alpha_{0}+\alpha_{1}X+\alpha_{2}Z+\varepsilon_{2}$$
(4)$$Y=\varphi_{0}+\varphi_{1}X+\varphi_{2}M+\varphi_{3}Z+\varepsilon_{3}$$
where Y represents the CES-D score of the elderly, X represents whether the elderly get a pension, and M represents intergenerational support. When $\beta_{1}$,$\;\alpha_{1}$, $\varphi_{2}$ are all significant, according to some research [55,56,57], there is a mediating impact. Furthermore, if $\varphi_{1}$ is not significant after the mediation effect is validated, it suggests the existence of a complete mediation effect; otherwise, it shows the existence of a partial mediation effect.

#### 3.3.3. Quantile Regression

Due to the spatiotemporal heterogeneity of variables, data may have outliers, resulting in non-normal distribution. However, conditional mean regression cannot rule out the influence of outliers, which may cause the estimation results to deviate. Quantile regression can divide the explained variable into different quantile levels according to the value range of the explained variable. The regression results are not only not interfered with by outliers, but also reflect the heterogeneity effects of the explanatory variable on the explained variable with different quantile levels in a more comprehensive way, making the regression results more accurate. In general, additive fixed-effect quantile models reduce the number of parameters to be estimated by assuming that individual effects are independent of quantiles. Powell [58] used the non-additive fixed effect to relax the assumption that the individual effect is independent of quantile and maintain the non-separable interference term in quantile estimation. The model is constructed as follows:(5)$$Y_{\tau }=X_{it}^{\prime }\beta \left(U_{it}^{*}\right)$$
where *X* represents explanatory variables and *U* represents individual abilities or tendencies and is an unknown form function of fixed effects and specific observed disturbance terms:(6)$$U_{it}^{*}=f\left(\alpha_{i},U_{it}^{\;}\right)$$
where $U_{it}^{\;}\sim U\left(0,1\right)$. Powell’s QRPD estimator allows the probability to vary between different individuals or even within individuals, as long as the fluctuation is orthogonal to the instruments. Thus, QRPD has a conditional and an unconditional restriction. In addition, the model can be estimated using the Markov Monte Carlo optimization method.
(7)$$\begin{array}{l} P\left(Y_{it}\leq X_{it}^{\prime }\beta \left(\tau \right)|X_{i}\right)=P\left(Y_{is}\leq X_{is}^{\prime }\beta \left(\tau \right)|X_{i}\right)\\ \;P\left(Y_{it}\leq X_{it}^{\prime }\beta \left(\tau \right)\right)=\tau \end{array}$$

## 4. Results

### 4.1. Descriptive Statistics

Table 1 gives the descriptive statistics of the variables in this paper. According to Table 1, we can see that the average depression level of the elderly is 13.764, indicating that the overall depression status of the elderly in China is relatively low. However, further combined with Figure 1, it can be seen that there are still a considerable number of elderly people with relatively high levels of depression. Observing the overall average of pensioners, the elderly who receive pension benefits only account for 60.7% of the total sample, indicating that there is still a large gap in the coverage of China’s pension schemes, and there are still many non-pensioners. Further, the mean differences between groups of variables are calculated by whether they receive pension benefits, finding that the depression degree of pensioners is 13.699, while the degree of depression in non-pensioners is 13.802. The depression levels of pensioners were significantly lower than those of non-pensioners. In addition, further observation shows that the average levels of the reverse intergenerational transfer amount, intergenerational life care, and intergenerational spiritual comfort of pensioners are also higher than those of non-pensioners. This result may indicate that receiving pension benefits can reduce the depression of the elderly, and receiving pension benefits may affect the depression of the elderly by affecting reverse intergenerational support, intergenerational life care, and intergenerational spiritual comfort. However, establishing the impact of receiving pension benefits on the mental health of the elderly and its mechanism requires rigorous empirical analysis in the following sections.

### 4.2. Basic Regression Results

In this paper, the stepwise regression method was used to test the influence of receiving pension benefits on the elderly’s mental health. Additionally, the regression results are shown in Table 2. Model (1) only contains the core variables and leaves out the control variable from the regression findings. Model 2 controls the individual characteristics of the elderly. Model 3 takes into account both individual and family characteristics. Additionally, Model 4 expanded upon Model 3 by including a time-point fixed effect. The regression coefficient of receiving pension benefits in model (1) is −0.207, which is significant at a significance level of 5%. In model (2), the regression coefficient of receiving pension benefits is −0.266, which is significant at the 1% significance level. In model (3), the regression coefficient of receiving pension benefits is −0.263, which is significant at the 1% significance level. In model (4), the regression coefficient of receiving pension benefits is −0.248, which is significant at the 1% significance level. Model (1), Model (2), Model (3), and Model (4) all have substantial negative regression coefficients for receiving pension benefits, demonstrating that receiving pension benefits is adversely connected with the degree of depression in the elderly. Receiving pension benefits can significantly reduce the degree of depression in the elderly and improve their mental health, and the conclusions obtained are somewhat robust. The elderly’s gender, marital status, education level, physical health status, and family income have a significant negative impact on the degree of depression in the elderly, according to the control variables.

### 4.3. Robustness Test

#### 4.3.1. Change the Variables

The simplified scale CES-D8 score was used to assess the mental health of the elderly in the previous study. To verify that the conclusion of this paper is not the result set by a specific topic, the explained variable was replaced by the complete scale CES-D20 score and re-regressed, as shown in Table 3 model (5). Furthermore, because the explanatory variable in the above analysis is the dummy variable of whether or not a person receives pension benefits, the differential impact of the pension amount on the mental health of the elderly is easily overlooked. However, the pension standards of each region are extremely different. Therefore, we replace the explanatory variable with the natural logarithm of the average amount of basic pensions for urban and rural residents in each province. The regression result is shown in Table 3 model (6). The analysis shows that the regression coefficients of the core explanatory variables in model (5) and model (6) are significantly negative, which indicates that the conclusions of this paper are robust to some extent.

#### 4.3.2. Controlling Macro Factors

Macro factors can influence whether the elderly can receive pension benefits and their mental health. However, the control variables in this paper were primarily characteristic variables at the individual and family levels of the elderly, and the influence of macro factors was not taken into account, as they could easily lead to estimation errors. However, including too many macro variables in the model would cause multicollinearity, and no matter how many macro variables were included, macro factors’ influence could not be completely controlled. To control the influence of macro factors from two levels of village and province, we referred to Yu [59]. We focused on the income level and the age structure of the population, which are two representative macro factors, in light of the unbalanced economic development and the basic national conditions of “getting old before getting rich.” The following are the specific methods: to begin with, the model incorporated the income level of village residents as well as the population age structure. Second, we incorporated each province’s income level and population age structure into the model. The results are shown in Table 4 model (7) and model (8). The pension regression coefficients in models (7) and (8) are both significantly negative, indicating that the conclusion of this paper is robust to some extent.

#### 4.3.3. Endogeneity Test

Due to a self-selection problem in the elderly’s basic pension, the elderly frequently choose whether or not to participate in social endowment insurance based on their risk preferences and family situation, which has a direct impact on whether the elderly can receive pension benefits, resulting in inaccurate estimation results. We analyzed this endogenous problem further and, following Pan [38], re-estimated the model using the received pension benefits rate of social endowment insurance in each village or community as an instrumental variable. The rate of receiving pension benefits is substantially linked with whether each elderly person receives pension benefits in theory, but the village or community’s rate of receiving pension benefits from social endowment insurance has no direct effect on the elderly’s mental health, which fits the condition of instrumental variable externality. Table 4 model (9) shows regression results utilizing the instrumental variable method. The regression coefficient of the instrumental variable is notably negative, demonstrating that the finding in this article is robust to some extent. Furthermore, Kleibergen–Paap rk LM and Kleibergen–Paap rk Wald F statistics are used to test whether the instrumental variable is identified and whether it is a weak instrumental variable. The test findings reject the null hypothesis that the instrumental variable is unidentifiable and weak at the 1% level, implying that the instrumental variable is effective.

### 4.4. Test of Mediating Effect

As has been previously stated, a variety of factors, such as intergenerational family support, can influence the mental health of older people. Pensions, as an important part of the social support mode, may have a “crowding in” or “crowding out” effect on intergenerational support, and elderly mental health cannot be separated from intergenerational support [48]. The stepwise method was used to see whether receiving pension benefits has an impact on the elderly’s mental health through changing intergenerational support. Table 4 displays the test results.

By analyzing model (10), model (11) and model (12) in Table 4, it can be seen that the regression coefficient of receiving pension benefits in model (10) is significantly positive, indicating that receiving pension benefits can reduce depression and improve mental health in the elderly. Receiving pension benefits has a significant positive regression coefficient in model (11), indicating that receiving pension benefits can encourage the elderly to provide more reverse intergenerational economic support for their children. The regression coefficient of the amount of reverse intergenerational support is positive in model (12), indicating that the amount of reverse intergenerational economic support is positively correlated with the elderly’s mental health. Increases in the amount of reverse intergenerational economic support paid by the elderly will exacerbate depression among the elderly, which is harmful to their mental health. After taking reverse intergenerational economic support into account, the total effect of the receipt of pension benefits on elderly mental health is −0.205 (0.318 × 0.038 − 0.217), which is lower than the effect of the receipt of pension benefits on elderly mental health in model (10). This finding suggests that receiving pension benefits may “crowd out” intergenerational economic support from children while “crowding in” intergenerational financial assistance for adult children, resulting in reverse intergenerational transfer and reducing the impact of the pension benefits receipt on elderly depression.

Using the same method to analyze models (10), (13), and (14), receiving pension benefits can directly reduce the depression status of the elderly, improve their mental health, and further improve the mental health of the elderly by increasing the intergenerational life care they receive. Using the same method to analyze models (10), (15), and (16), receiving pension benefits can not only directly reduce the depression of the elderly, but also further reduce the degree of depression and improve the mental health of the elderly by increasing the intergenerational spiritual comfort provided by children to the elderly.

Through the above analysis, it can be found that receiving pension benefits may “crowd out” the amount of intergenerational economic support from children, “crowd in” the elderly’s financial help for adult children, leading to the reverse intergenerational transfer, and reducing the effect of the receiving pension benefits on depression of the elderly. However, as the elderly share pensions with their children, there is an increase in the provision of care and emotional comfort by their children. Overall, receiving pension benefits improves the mental health of the elderly.

### 4.5. Heterogeneity Analysis

#### 4.5.1. Heterogeneity Analysis of Different Groups

We placed all of the elderly respondents into a single homogeneous group in the preceding studies. However, there may be heterogeneity in the policy effect of pensions on mental health for the elderly with varied characteristics. We grouped the elderly according to the following criteria to investigate the heterogeneous impact of the receipt of pension benefits on the mental health of the elderly with various characteristics: (1) the elderly were split into the older elderly and the younger elderly, with 75 years as the critical value, according to their age classification; (2) the elderly were classified as male or female based on their gender; (3) the elderly were classified as living with their children or living alone based on their residential pattern; and (4) the elderly were classified as residing in urban or rural locations based on their household registration.

Table 5 models (17) and (18) show the impact of receiving pension benefits on the mental health of the younger elderly and older elderly, respectively. The regression coefficient of receiving pension benefits in model (17) is not significant, while the regression coefficient of receiving pension benefits in model (18) is significantly negative, indicating that receiving pension benefits can significantly reduce depression and improve mental health in the older elderly, but it has no significant impact on the mental health of the younger elderly. The marginal effect of receiving pension benefits on the older elderly’s mental health is considerably greater. The Fischer combination test is employed for 1000 samples to generate the empirical distribution of group differences to examine the significance of the differences in the coefficients obtained by group regression. The empirical distribution results show that in the regression of different age categories, the marginal effect of receiving pension benefits rejects the null hypothesis that there is no difference at the 1% significance level. This demonstrates that the marginal effect of receiving pension benefits for the younger elderly and the older elderly differs significantly. Table 5 models (19) and (20) show the impact of receiving pension benefits on men’s and women’s mental health, respectively. The regression coefficient of receiving pension benefits in model (19) is not significant, while the regression coefficient of receiving pension benefits in model (20) is significantly negative. This finding suggests that receiving pension benefits can significantly reduce women’s melancholy and enhance their mental health, but the effect on men’s mental health is minor and the marginal effect on women’s mental health is greater. This finding is supported by the Fischer combinatorial test.

Table 5 models (21) and (22) show the impact of receiving pension benefits on the mental health of the elderly with various living arrangements. The regression coefficient of receiving pension benefits in model (21) is significantly negative, whereas the regression coefficient of receiving pension benefits in model (22) is not significant. This shows that receiving pension benefits can significantly reduce depression in the elderly who live with their children and improve their mental health level, while the impact on the mental health of the elderly who live alone is small. The results show that the marginal effect of receiving pension benefits on the mental health of elderly people who live with children is stronger, as Fisher’s combination test demonstrates. Table 5 models (23) and (24) show the impact of receiving pension benefits on the mental health of the elderly in urban and rural locations, respectively. The regression coefficients of receiving pension benefits in models (23) and (24) are significantly negative, demonstrating that receiving pension benefits can greatly reduce depression and improve mental health in urban and rural older people. Further observation reveals that the regression coefficient of pension benefits received in model (23) is lower than in model (24), implying that receiving pension benefits has a higher marginal effect on depression among the rural elderly, which is supported by Fisher’s combinative test.

#### 4.5.2. Panel Quantile Regression

The quantile model divides the elderly into different quantile levels according to the value range of their mental health status, which can more comprehensively reflect the heterogeneity of receiving pension benefits for the elderly with different mental health statuses than the mean regression of the conditional expectation model. In this paper, nine representative integer quantiles from 10% to 90% were selected, and the panel quantile model based on Markov Monte Carlo was used for regression. The regression coefficients for the mental health of the elderly receiving pension benefits at different quantile levels are obtained. The results are shown in Figure 2.

Receiving pension benefits has varied effects on the elderly with different mental health states, according to Figure 2. The regression coefficient for pension tends to be 0 at the lower quantile level. The findings imply that receiving pension benefits does not lower depression or improve mental health in the elderly with relatively low levels of depression and excellent mental health. The regression coefficient for receiving pension benefits is significantly negative at the 1% significance level at the mid-to-high quantiles from 30% to 90%. This shows that receiving pension benefits can significantly reduce depression and improve mental health in the elderly who have a high level of depression and poor mental health. Further observation of the regression coefficient of receiving pension benefits reveals that as the quantile level rises, the absolute value of the regression coefficient of receiving pension benefits rises as well. This finding suggests that the higher the level of depression and the worse the elderly person’s mental health, the greater the marginal effect of receiving pension benefits. The marginal effect of receiving pension benefits is stronger for elderly people with poor mental health.

## 5. Discussion

In general, generation A looks after generation B, and generation B looks after generation C in Western countries, which is a one-way intergenerational relationship in the “relay mode.” China’s intergenerational relationship, unlike that of Western countries, is a “feedback mechanism” of raising and supporting [47]. Parents and children are inextricably linked, and having children is one of the most essential ways for Chinese people to effectively withstand the hazards of old age. The elderly tend to have a stronger emotional attachment to their children [60]. Children’s intergenerational support will have a direct impact on the elderly’s emotional and mental health. However, with a rapidly declining fertility rate and an escalating aging trend, along with the continuous miniaturization and nucleation of China’s family structure, the function of the family as the primary support mode for the elderly [61] has waned, but the pension benefits represented by social endowment insurance have progressively evolved and become an important part of the elderly’s support system. However, according to the researchers, China’s social endowment insurance system is still in its early stages of growth, with social endowment insurance coverage being limited. The level of pension benefits is often inadequate, unable to meet the fundamental needs of the elderly, and pension benefits may “crowd out” children’s intergenerational economic support to some extent, reducing the elderly’s wellbeing [9,41].

We used data from the 2012, 2016, and 2018 Chinese Family Panel Studies (CFPS) to assess the mental health of the elderly using the CES-D8 Depression Scale to investigate whether China’s pension schemes may improve their mental health. The effect of receiving pension benefits on the mental health of the elderly was investigated using a two-way fixed effects model and so on. Receiving pension benefits, according to the findings, can significantly reduce depression in the elderly and improve their mental health. The results remained stable after modifying the explanatory variables, controlling for macro factors, and accounting for the endogenous problem. Pensions, as an integral aspect of the social support mode, can affect the mental health of the elderly not only directly, but also indirectly through the influence of family intergenerational support. Receiving pension benefits is observed to “crowd out” children’s intergenerational economic support and “crowd in” the elderly’s economic assistance to children, resulting in the reverse intergenerational transfer of pensions and reducing the policy effect of pensions on the elderly’s mental health. There is no such significant resource for China’s elderly population, who have basically passed over most of their homes and land assets to their married sons. If they help their children financially, it can lead to overtaking, which lowers their quality of life and hurts their health [62].

Although receiving pension benefits can cause reverse intergenerational transfer, to a certain extent, it can improve the relative economic status of the elderly inside and outside the family, reduce the economic dependence of the elderly on their children, reduce the pressure on children’s support, and alleviate the conflict between children’s “career” roles and “filial children” roles [44]. When receiving pension benefits causes reverse intergenerational transfer, the spiritual comfort and emotional support provided by children gradually increase. In general, receiving pension benefits improves the mental health of the elderly. Furthermore, we discovered that the effect of the receipt of a pension on the elderly with various characteristics is heterogeneous, with the marginal effect being stronger for the elderly, women, the elderly living with their children, and the elderly in rural areas. Generally speaking, the elderly, women, and the elderly in rural areas are relatively disadvantaged, have relatively low-income levels, and have relatively poor mental health [63,64]. Pensions may greatly help them to build confidence and reduce their worries about later life, thus benefiting their mental health.

There were various limitations in our research. First, the information in this article was gathered through questionnaires, which may exaggerate the strength of the link between mental health and certain variables [65]. Second, the questions in the database’s version of the CES-D scale only included the frequency of psychological problems in the previous week, which could lead to measurement errors. For example, if the respondents had received a shock in the previous week (such as the death of a family member, etc., they would almost certainly have experienced abnormal mood swings). Finally, we only looked at the impact of whether or not the elderly receive pension benefits on their mental health, but due to survey data limitations, it is difficult to determine the impact of pension amount on mental health. We, however, go even further. The test results show that the elderly pension amount and depression status have significant negative effects and that increasing the number of pensions can effectively alleviate the elderly’s depression status and improve the level of mental health in the elderly.

## 6. Conclusions

Based on data from the 2012, 2016, and 2018 Chinese Family Panel Studies (CFPS), our study empirically examined the effect of the receipt of pension benefits on the elderly’s mental health status and explored whether pensions could in turn affect the elderly’s mental health status through intergenerational support, and the heterogeneous effect of pension benefits receipt on different elderly. We found that: (1) receiving pension benefits can help the elderly by reducing depression and improving their mental health. (2) While receiving pension benefits can boost the elderly’s reverse intergenerational economic support provided to adult children to some extent, the policy effect of pension benefits on the elderly’s mental health is lessened to some extent. However, as the elderly share their pensions with their children, their children are more likely to provide care and emotional support. Pensions boost the mental health of the elderly in general. (3) The elderly with varied characteristics have different reactions to receiving pension benefits. The older elderly, women, the elderly living with their children, and the elderly in rural areas all benefit more from receiving pension benefits. Furthermore, this study discovered that the marginal effect of receiving pension benefits on mental health is stronger for the elderly with poor mental health.

Based on the aforementioned conclusions, we recommend the following strategies to actively respond to aging and develop a healthy China: first, we will focus more on enhancing pension benefits for the elderly. In light of dwindling intergenerational support, social endowment insurance should serve as a “safety net” for the elderly. It is recommended that relevant departments improve the pension adjustment mechanism, appropriately increase the basic pension for the elderly by the age of the elderly population and the nature of household registration, effectively avoid the income risk brought on by age growth, and avoid the elderly’s psychological sense of deprivation and abandonment. Second, the establishment of a diverse pension service system improves the elderly’s life satisfaction. It is recommended that relevant departments increase investment in elderly care services, promote the development of a diverse pattern of elderly care services, vigorously develop inclusive elderly care services, provide health care, assisted meals, and baths for the elderly, help them engage in spiritual and cultural activities, enrich the daily lives of the elderly, and effectively improve the elderly’s life satisfaction. Finally, fine-tune and perfect the fundamental social endowment insurance operation mode. Residents can voluntarily choose the level of contribution under the relevant policies of social endowment insurance, but residents, particularly women, rural residents, and other vulnerable groups, generally choose a lower level of contribution based on their income, which has a significant impact on their pension benefits. As a result, it is recommended that relevant departments provide premium subsidies to women and rural residents and improve the pension benefits level of disadvantaged groups, thereby reducing depression and improving mental health.

## Figures and Tables

**Figure 1 ijerph-19-08721-f001:**
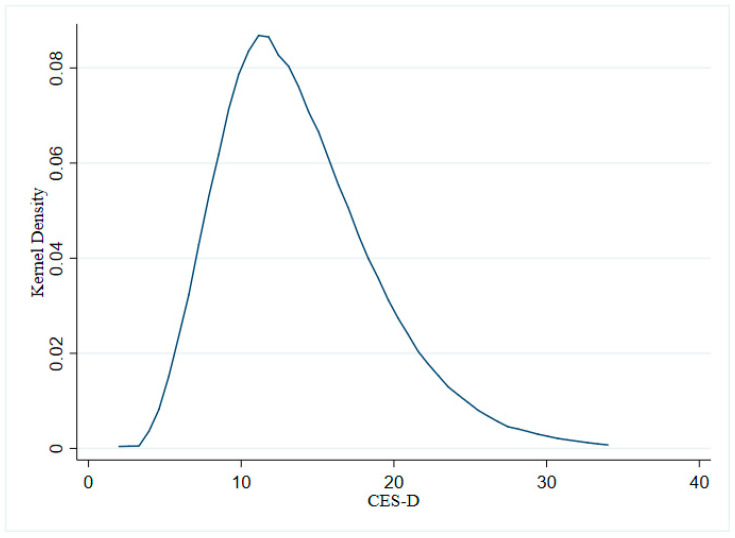
Kernel density curve of CES-D.

**Figure 2 ijerph-19-08721-f002:**
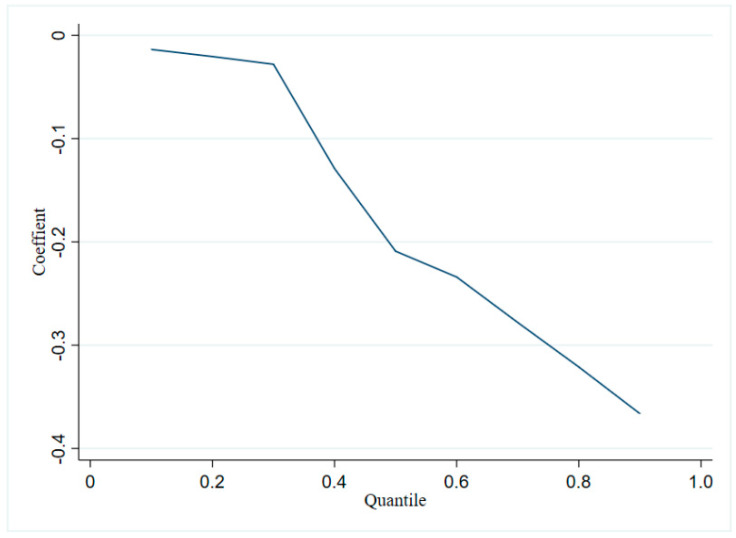
Quantile regression results.

**Table 1 ijerph-19-08721-t001:** Descriptive statistics.

Variables	Variable Description	Full Sample	Pensioners	Non-Pensioners	Diff
		Mean	Mean	Mean	Mean
CES-D	the score of CES-D scale	13.764	13.699	13.802	−0.103 ***
Pension	1 = yes, 0 = no	0.607			
RLS	Natural logarithm of the amount of reverse intergenerational economic support	2.614	2.248	2.851	−0.603 ***
ILC	Frequency of children providing life care to the elderly	0.506	1.36	1.669	−0.309 ***
ISC	The closeness of the relationship between children and the elderly	3.742	3.482	3.91	−0.428 ***
Age	age^2^/100	49.402	49.654	49.24	0.414 ***
Gen	1 = male, 0 = female	0.496	0.524	0.478	0.046 ***
Mar	1 = married, 0 = unmarried	0.825	0.83	0.821	0.009
Edu	Number of years of education received	5.039	5.297	4.873	0.424 ***
Hea	1 = Great/Good, 0 = Not Bad/Bad/Worse	0.673	0.665	0.679	−0.015
Pinc	The average income of the household members	9.105	9.212	9.036	0.176 ***
Size	Number of people in the household	3.279	3.268	3.287	−0.019
Urn	1 = urban, 0 = countryside	0.474	0.538	0.432	0.106 ***

Note: *** represents the significance levels of 1%.

**Table 2 ijerph-19-08721-t002:** Basic regression results.

Variables	(1)		(2)		(3)		(4)	
Pension	0.207 **	(0.093)	−0.266 ***	(0.090)	−0.263 ***	(0.090)	−0.248 ***	(0.090)
Age			0.022 *	(0.009)	0.025 **	(0.009)	−0.006	(0.012)
Gen			−1.096 ***	(0.118)	−1.097 ***	(0.118)	−1.021 ***	(0.120)
Mar			−1.335 ***	(0.236)	−1.316 ***	(0.238)	−1.435 ***	(0.239)
Edu			−0.032 **	(0.015)	−0.035 *	(0.015)	−0.027 *	(0.016)
Hea			−1.708 ***	(0.108)	−1.706 ***	(0.108)	−1.698 ***	(0.109)
Pinc					−0.062	(0.049)	−0.118 **	(0.051)
Siz					−0.014	(0.044)	0.007	(0.044)
Urn					0.123	(0.260)	0.045	(0.261)
Constant	13.929 ***	(0.066)	15.019 ***	(0.546)	16.258 ***	(0.651)	18.272 ***	(0.824)
Individual-fixed effect	YES		YES		YES		YES	
Year-fixed effects							YES	
Observations	9672		9672		9672		9672	

Note: The brackets represent t values, with ***, **, and * representing the significance levels of 1%, 5%, and 10%.

**Table 3 ijerph-19-08721-t003:** Robustness test results.

Model	(5)	(6)	(7)	(8)	(9)
	Change Variables		Control Macro Factors		Endogeneity Test
Explanatory variable	−0.793 **	−0.242 *	−0.216 *	−0.194 **	−0.230 ***
	(0.398)	(0.146)	(0.127)	(0.091)	(0.139)
Control variables	YES	YES	YES	YES	YES
Inc			−0.086 *	−2.963 **	
			(0.091)	(1.289)	
Pop			−1.667 ***	−0.004	
			(0.507)	(0.003)	
Constants	41.348 ***	19.927 *	20.817 ***	51.210 ***	22.847 ***
	(15.277)	(1.354)	(0.898)	(12.140)	(0.457)
Individual-fixed effects	YES	YES	YES	YES	
Year-fixed effects	YES	YES	YES	YES	
Observations	7771	9672	5714	9672	9672

Note: The brackets represent t values, with ***, **, and * representing the significance levels of 1%, 5%, and 10%.

**Table 4 ijerph-19-08721-t004:** Mechanism test results.

Model	(10)	(11)	(12)	(13)	(14)	(15)	(16)
Variables	CES-D	RIS	CES-D	ILC	CES-D	ISC	CES-D
Mediating variable			0.038 *		−0.044 ***		−0.148 ***
			(0.022)		(0.015)		(0.026)
Pension	−0.219 **	0.318 ***	−0.217 **	0.530 ***	−0.196 **	0.426 ***	−0.156 ***
	(0.090)	(0.042)	(0.092)	(0.063)	(0.091)	(0.035)	(0.091)
Control variables	YES	YES	YES	YES	YES	YES	YES
Constants	22.838 ***	−1.183	22.852 ***	4.937 ***	23.056 ***	3.716 ***	23.389 ***
	(0.432)	(0.200)	(0.438)	(0.230)	(0438)	(0.166)	(0.443)
Observations	9672	9672	9672	9672	9672	9672	9672

Note: The brackets represent t values, ***, **, and * representing the significance levels of 1%, 5%, and 10%.

**Table 5 ijerph-19-08721-t005:** Heterogeneity test results.

Group	(17)	(18)	(19)	(20)	(21)	(22)	(23)	(24)
	Younger Elderly	Older Elderly	Men	Woman	Living with Children	Living Alone	Urban	Rural
Pension	−0.155	−0.463 **	0.080	−0.335 **	−0.477 **	−0.063	−0.064 **	−0.314 **
	(0.109)	(0.204)	(0.121)	(0.136)	(0.152)	(0.176)	(0.134)	(0.124)
Control variables	YES	YES	YES	YES	YES	YES	YES	YES
Constants	17.595 ***	14.248 ***	22.230 ***	23.407 ***	16.862 ***	13.489 ***	22.435 ***	23.571 ***
	(1.500)	(3.628)	(0.665)	(0.705)	(1.303)	(6.551)	(0.700)	(0.706)
Observations	7143	1808	4801	4870	4285	3086	5088	4583

Note: The brackets represent t values, *** and **representing the significance levels of 1% and 5%.

## Data Availability

Not applicable.

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
