# Peer review of "The Impact of Social Pension Schemes on the Mental Health of the Chinese Elderly: A Mediating Effect Perspective of Two-Way Intergenerational Support"

_ijerph, 2022, doi:10.3390/ijerph19148721_

Round 1
Reviewer 1 Report
I found the paper interesting and original. The research performed on a very large sample allowed the Authors to reveal that receiving pension benefits improves the mental health of the elderly. It also turned out that the effect of pensions on the elderly with different characteristics is heterogeneous. The conslusions were elaborated with the use of advanced statistical methods like fixed effects model, mediating effect model, quantile regression.
In general, text is on a high level of scientific work. In my opinion design, structure, used methods and discussion of results are prepared and realized very well.
I have only one small concern - I did not find the aim of the research stated (should be placed in summary or introduction)
Author Response
There appears to be a flaw in the system,as the incorrect file cannot be deleted. Please download the file entitled "Response to Reviewer 1 Comments", which is the correct file.

Reviewer 2 Report
The research deals with a very interesting topic, namely the impact of social pension schemes on the mental health of the Chinese elderly population. The article is well structured, well written and meets the requirements of a scientific publication. The methodology is adequate, well explained, it would have been more interesting to extend the research over a longer period of time, the results are relevant, the conclusions are objective. The bibliography is adequate and sufficient for the research carried out.
Reviewer 3 Report
Reviewer report on “The Impact of Social Pension Schemes on Mental Health of the Chinese Elderly” submitted to IJERPH
The paper under review examines an important question that is of both academic interest and policy relevance, that of how social pension programs affect the elderly’s mental health status (measured by their depression scale) in China, using panel data from the CFPS. The paper is, in general, well-written. The literature review is extensive and solid. Much attention was paid to modeling issues. Overall, the quality of this paper is much higher than most of the papers I have reviewed for the IJERPH. That being said, I note several areas in the paper that may be improved further. My more-detailed comments are outlined below, in no particular order.
1. The first paragraph mentioned many numbers taken from some “reports.” These reports deserved to be formally cited in the text and added to the references list.
2. More background information should be provided on the pension program.
3. On sample restrictions. The paper mentions on page 5, line 219, that “we kept samples of women over 50 and men over 60, according to China’s current legal retirement age”. This might not be accurate, as China’s legal retirement age for women is 55, not 50, and that only applies to those working in urban sectors, but not rural residents.
4. related to the previous point: is the final analytical sample a balanced panel? If it is not a balanced panel, then the fixed effects specification might not be as powerful as it appears.
5. Descriptions of the mediating variables seem unclear. For example, how is “intergenerational spirit comfort” measured exactly? For example, is it a counious or discrete variable?
6. Shouldn’t the error terms in equations (1)-(4) be different? There are now all denoted as epsilon_it, but they intend to capture different things, right?
7. I believe that the main citation for the models (2)-(4), Wen, Z et al. (and the source of your IV method as well) is a Chinese paper. For an international journal, the source of your key method should be more internationally recognizable.
8. It would be very helpful to state clearly your identifying assumptions. That way, the reader will be able to assess the validity of your results.
9. Some statements seem incorrect/unclear. For example, line 293 on page 6: “…, conditional mean regression cannot capture and identify these outliers,...” In fact, it is because conditional mean regression captures the influence of outliers that it may yield misleading estimates – put differently, condition mean regression is not immune to the problem of outliers. Also, measures such as Cook’s distance can be used to detect outliers.
10. I am concerned about the results of the IV estimation – the point estimate of the IV regression is more than 100 times those of linear regressions in most of the cases!! This deserves some scrutiny. For example, if an estimate of 13 is considered consistent (-recall the concept of consistency), can an estimate of 0.13 be considered consistent?
11. Typos and grammatical errors can be spotted at times. For example, line 118 on page 3: the first sentence: ”Second, is the impact of……” seems grammatically incorrect. Line 200 on page 4: “…will affects…” should be “will affect”. The authors should check the text more carefully to spot and correct all remaining errors.
Round 2
Reviewer 3 Report
The manuscript has been much improved, although the authors' response to my point 4 on "unbalanced panel" seems problematic -- if an unbalanced panel is not a problem, then why would people talk about "attrition bias"? At the minimum, the "fixed effects" i.e., individual-specific dummies, would not be completely differenced out if the panel is not balanced. Think extremely, what if the observations are missing in a way such that there is NO repeated observation at all (i.e. each person shows up only once in the dataset) -- in this case, the panel model becomes a pooled OLS model. I am sure Chen Qiang, whoever this person is, will point out the difference between a pooled OLS model and a fixed-effect model (see also Greene, 2018, for more detail). For the results of using a balanced panel and an unbalanced panel to be the same, you would need strong assumptions.
References
Greene, W. 2018. Econometric Analysis. Pearson.
